# Organ preservation solution containing dissolved hydrogen gas from a hydrogen-absorbing alloy canister improves function of transplanted ischemic kidneys in miniature pigs

Eiji Kobayashi[1,2,3], Motoaki Sano[1,3]*

**1** Department of Cardiology, Keio University School of Medicine, Tokyo, Japan, **2** Department of Organ Fabrication, Keio University School of Medicine, Tokyo, Japan, **3** Center for Molecular Hydrogen Medicine, Keio University, Tokyo, Japan

\* msano@a8.keio.jp

**Data Availability Statement:** All relevant data are within the paper and its Supporting Information files.

## Abstract

Various methods have been devised to dissolve hydrogen gas in organ preservation solutions, including use of a hydrogen gas cylinder, electrolysis, or a hydrogen-generating agent. However, these methods require considerable time and effort for preparation. We investigated a practical technique for rapidly dissolving hydrogen gas in organ preservation solutions by using a canister containing hydrogen-absorbing alloy. The efficacy of hydrogen-containing organ preservation solution created by this method was tested in a miniature pig model of kidney transplantation from donors with circulatory arrest. The time required for dissolution of hydrogen gas was only 2–3 minutes. When hydrogen gas was infused into a bag containing cold ETK organ preservation solution at a pressure of 0.06 MPa and the bag was subsequently opened to the air, the dissolved hydrogen concentration remained at 1.0 mg/L or more for 4 hours. After warm ischemic injury was induced by circulatory arrest for 30 minutes, donor kidneys were harvested and perfused for 5 minutes with hydrogen-containing cold ETK solution or hydrogen-free cold ETK solution. The perfusion rate was faster from the initial stage with hydrogen-containing cold ETK solution than with hydrogen-free ETK solution. After storage of the kidney in hydrogen-free preservation solution for 1 hour before transplantation, no urine production was observed and blood flow was not detected in the transplanted kidney at sacrifice on postoperative day 6. In contrast, after storage in hydrogen-containing preservation solution for either 1 or 4 hours, urine was detected in the bladder and blood flow was confirmed in the transplanted kidney. This method of dissolving hydrogen gas in organ preservation solution is a practical technique for potentially converting damaged organs to transplantable organs that can be used safely in any clinical setting where organs are removed from donors.

**Funding:** This study was funded by Doctors Man Co., Ltd. (to Eiji Kobayashi). This funder had no role in study design, data collection and analysis, decision to publish, or preparation of the manuscript.

**Competing interests:** Eiji Kobayashi received a research grant from Doctors Man Co., Ltd.. Keio University and Doctors Man Co., Ltd. have jointly applied for a patent (patent no. 2019-081425). Eiji Kobayashi and Motoaki Sano are the inventors of this patent. This does not alter our adherence to PLOS ONE policies on sharing data and materials.

## Introduction

Transplantation of marginal organs from donors after circulatory arrest is an important option to reduce the waiting period for recipients. Progress has been made in perfusion of donor organs with diluted donor blood [1], but there are few methods that can be easily used at many sites where donors may be found.

Hydrogen gas has been shown to have various biological effects, including suppression of ischemia-reperfusion injury in animal studies [2–5]. Ischemia-reperfusion injury is an inevitable complication of solid organ transplantation and limiting this type of injury can increase graft survival. Use of hydrogen gas has been reported to be effective in transplantation models of various organs, including the small intestine [6–7], lung [8–14], liver [15–18], heart [19, 20], osteochondral tissue [21], and kidney [22]. It is possible to expose the excised organ to hydrogen gas ex vivo without the donor and/or recipient inhaling the gas, and various methods have been devised to dissolve hydrogen gas in organ preservation solutions, including use of a hydrogen gas cylinder [6], electrolysis [18, 20, 21, 22], or a hydrogen-generating agent [17]. However, these methods require bulky equipment and dangerous high-pressure cylinders with strict regulations for handling, resulting in the need to expend considerable time and effort for preparation. Therefore, it is probably unrealistic to attempt the introduction of such methods into the clinical setting. Accordingly, a simple technique for rapidly dissolving hydrogen gas in organ preservation solutions is required.

A hydrogen-absorbing alloy is a compound that absorbs hydrogen when it is cooled or pressurized and then releases hydrogen when it is heated or depressurized. A hydrogen-absorbing alloy canister is filled with such an alloy, and these canisters have been used to supply hydrogen for fuel cells. We have developed a method of using a hydrogen-absorbing alloy canister to rapidly and conveniently dissolve hydrogen in organ preservation solutions at high concentrations. The canister storing hydrogen can be safely transported anywhere and can be easily connected to a bag containing conventional organ preservation solution, allowing hydrogen to be dissolved in organ preservation solution within a few minutes at the site of donor organ harvesting.

In the present study, the efficacy and safety of cold organ preservation solution containing hydrogen dissolved by this method were tested in a miniature pig model of kidney transplantation from donors with circulatory arrest. Previous studies were performed in juvenile domestic pigs, but we used miniature pigs to more closely reflect the clinical setting [23]. After circulatory arrest for 30 minutes, kidneys were harvested from the donor, flushed out, and stored in either hydrogen gas-containing organ preservation solution or conventional organ preservation solution. Then early kidney function after transplantation was compared between the two methods of preservation.

## Materials and methods

### Miniature pigs

Conventionally, animal studies on kidney transplantation have been performed using juvenile domestic pigs (about 4 months old). However, the donors and recipients were adult miniature pigs in the present study because this was considered to more closely correspond to the clinical setting. Miniature pigs do not exceed 30 Kg in weight when fully grown after 2 years. We used female pigs aged 25–40 months and weighing 20–26 Kg, purchased from Fujimicra Ltd., Shizuoka, Japan. Animals were treated in accordance with the Animal (Scientific Procedures) Protection Act 1986 of the United Kingdom. The pigs were housed in cages under temperature and light-controlled conditions (12-hour light/dark cycle) and were provided with food and

water ad libitum. The pigs were fasted for 12 hours prior to surgery with free access to water. Sedation with a mixture of midazolam/medetomidine/butorphanol was followed by endotracheal intubation and mechanical ventilation. Anesthesia was maintained with inhalational isoflurane. Midazolam and medetomidine were added according to the depth of anesthesia. Buprenorphine was administered as an intraoperative analgesic management. After intravenous administration of pentobarbital, saturated potassium chloride was rapidly administered intravenously to euthanize. Three animals were used to examine the flushing effect of the hydrogen-containing organ preservation solution and 9 animals were used for the organ transplantation experiment (one donor and two recipients were used for one set of experiments; n = 3). These experiments were conducted with the approval of the Research Council and Animal Care and Use Committee of Keio University [approval no: 16072-(1)]. Surgery was performed by a surgeon with over 200 clinical transplant operations, who is a steering member of the transplantation society and a permanent director of the transplantation society of Japan (E.K.).

### Hydrogen gas dissolution system based on a hydrogen-absorbing alloy canister

The hydrogen-absorbing alloy canister (Japan Steel Works, Ltd.) was a cylinder containing a hydrogen-absorbing alloy that can reversibly absorb and release hydrogen (Fig 1A). Since the internal pressure of the canister does not exceed 1 MPa, it does not correspond to high-pressure gas according to the Japan High-Pressure Gas Safety Act. The device for transferring hydrogen was assembled just before use and hydrogen was infused into the organ preservation solution from the hydrogen-absorbing alloy canister via a pressure regulating valve (Fig 1B and 1C). When the gauge pressure reached 0.06 Mpa, the plastic bag containing the organ preservation solution was disconnected from the pressure regulating valve. After vigorous manual shaking for 30 seconds or more, the bag containing the organ preservation solution was placed on ice and opened to the atmosphere. Then the concentration of hydrogen gas in the solution was measured over time by gas chromatography (TRIlyzer mBA-3000; Taiyo Co., Ltd., Kochi, Japan).

### Investigation of the effectiveness of hydrogen-containing preservation solution for flushing kidneys from donor pigs with circulatory arrest

A midline upper abdominal incision was made under general anesthesia, and the left and right kidneys were dissected. Then the chest was opened and the thoracic aorta was clamped to induce ischemia of all abdominal organs. After 30 minutes of warm ischemia, the kidneys were removed for use as donor organs. The donor animal was not treated with an anticoagulant such as heparin. The excised kidneys were immediately placed on the back table, divided into the right and left kidneys, and perfused for 5 minutes by gravity feed from a height of 1 m. One kidney was perfused with hydrogen-containing cold extracellular-type trehalose-containing Kyoto (ETK) solution and the other was perfused with hydrogen-free cold ETK solution. The number of drops of the solution per minute was counted for each kidney.

### Evaluation in a kidney transplantation model

After flushing with either hydrogen-containing cold ETK solution or hydrogen-free cold ETK solution for 5 minutes, the harvested kidneys were immersed and stored in the same ETK solution for 1 to 4 hours until transplantation into the recipient.

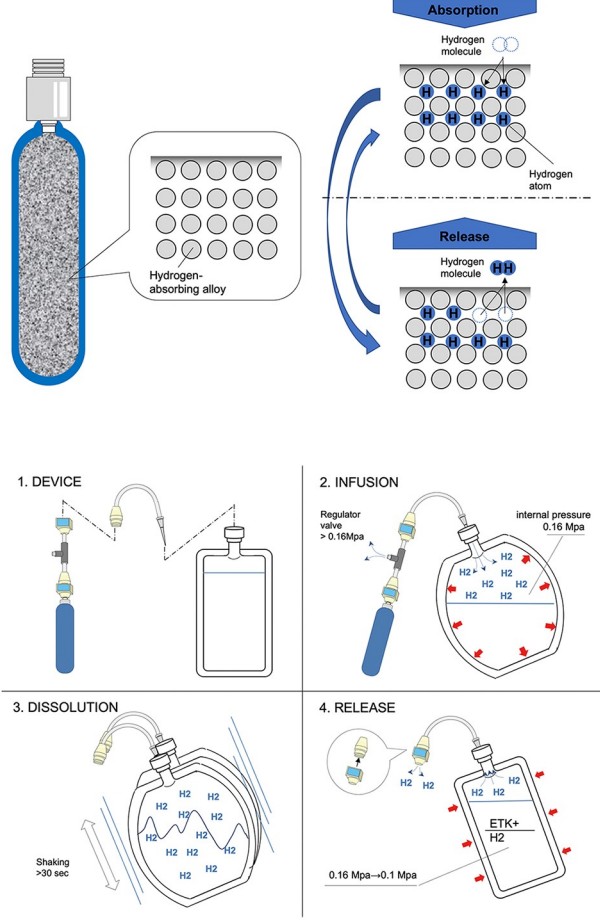

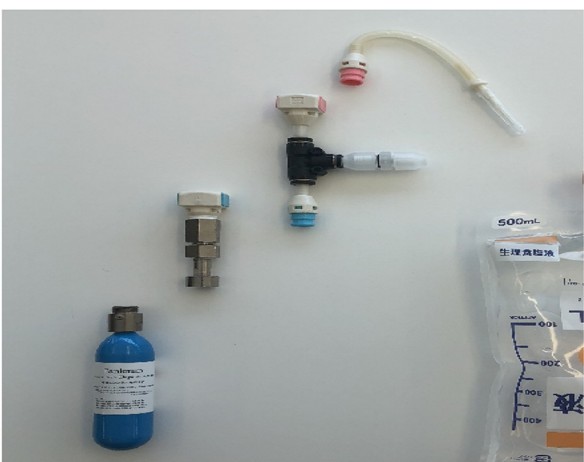

**Fig 1. Novel method of dissolving hydrogen gas in organ preservation solution by using a hydrogen-absorbing alloy canister.** (A) Mechanism of the hydrogen absorbing alloy. (B) Process of dissolving hydrogen in the organ preservation solution by using the hydrogen-absorbing alloy canister. (1) A hydrogen-absorbing alloy canister and a pressure regulating valve are connected to a bag of organ preservation solution. (2) Hydrogen gas is injected through the rubber stopper using a bottle needle. When the pressure reaches 0.06 Mpa, the bag of organ preservation solution is disconnected from the pressure regulating valve and hydrogen-absorbing alloy canister. (3) The organ preservation solution in the plastic bag is shaken vigorously by hand for more than 30 seconds. (4) The bag is opened to the atmosphere. (C) Components of the device for infusing hydrogen gas into organ preservation solutions.

Under general anesthesia, a midline abdominal incision was made to expose the recipient's left renal artery and vein [24]. After intravenous administration of 1 ml of heparin, the abdominal aorta was clamped immediately above the renal artery bifurcation. Then the renal vein was clamped with bulldog forceps, and the left kidney was resected from the renal artery leaving a Carrel patch configuration. Next, the stored donor kidney was anastomosed end-to-end with 5–0 nylon to the renal artery at the Carrel patch site. After arterial anastomosis, the peripheral renal artery was clipped again before unclamping the abdominal aorta. The total aortic clamp time was 30 minutes. Subsequently, the renal veins were joined by end-to-end continuous anastomosis with 6–0 nylon, and finally the ureter was anastomosed with 6–0 nylon knotted sutures. The transplanted kidney was wrapped in a thermal bag, and the temperature was kept at 20 ˚C or less by appropriate infusion of cold ETK solution [25]. Warm ischemic time was set at 1 hour. After reperfusion, blood flow was confirmed in the transplanted kidney. Then the right native kidney was excised and the abdomen was closed. Cefazolin sodium (25mg/kg) was given on the first postoperative day, and the animals were allowed free access to both food and water. The animals were observed for up to 6 days after surgery, and then were sacrificed for laboratory tests, urinalysis, and histopathological examination of the transplanted kidneys. Blood urea nitrogen (BUN) and creatinine (CRE) were measured in peripheral blood and urine samples by the urease-GLDH method and enzymatic method, respectively. Total urinary protein was measured by the pyrogallol red technique and urinary electrolytes were measured by an ion-selective electrode method, while inorganic phosphate (IP) was determined by an enzymatic method and glucose by the hexokinase/glucose-6-phosphate dehydrogenase method. The transplanted kidneys were fixed in 10% neutral buffered formaldehyde solution and cut in the longitudinal direction to include the papilla centering on the cortex. After embedding by a conventional method, thin sections were prepared and were stained with hematoxylin and eosin (H. E.) and Elastica van Gieson (EVG) stain. Histological examination was done by blinded pathologists and renal graft pathology was assessed according to the Banff classification [26, 27].

## Results

### Hydrogen concentration profile in organ preservation solutions

The time required for infusion of hydrogen gas was 2–3 minutes. Changes of the hydrogen concentration in three organ preservation solutions after infusion of hydrogen gas are shown in Fig 2. At the same pressure, UW solution and HTK solution had higher hydrogen concentrations than ETK solution. The hydrogen concentration declined very slowly at 4 ˚C under normal pressure and was 1 mg/L or more after 4 hours in all of the solutions. As ETK solution does not require strict temperature control for kidney preservation [28, 29], subsequent experiments on flushing and preservation of donor kidneys were performed using this solution.

### Flushing effect of hydrogen-containing organ preservation solution

Transplantation of marginal donor organs was performed to assess the effectiveness of cold hydrogen-containing ETK solution generated by our hydrogen gas infusion method. We used miniature pigs (25–40 months old) to more closely reflect the clinical transplantation setting. After warm ischemic injury was created by circulatory arrest for 30 minutes, the donor kidneys were harvested and perfused for 5 minutes with hydrogen-containing cold ETK solution or hydrogen-free cold ETK solution (Fig 3A). When hydrogen-containing cold ETK solution was used, the perfusion rate was faster from the initial stage of perfusion than with hydrogen-free ETK solution (Table 1). At the end of perfusion, visual inspection indicated that perfusion

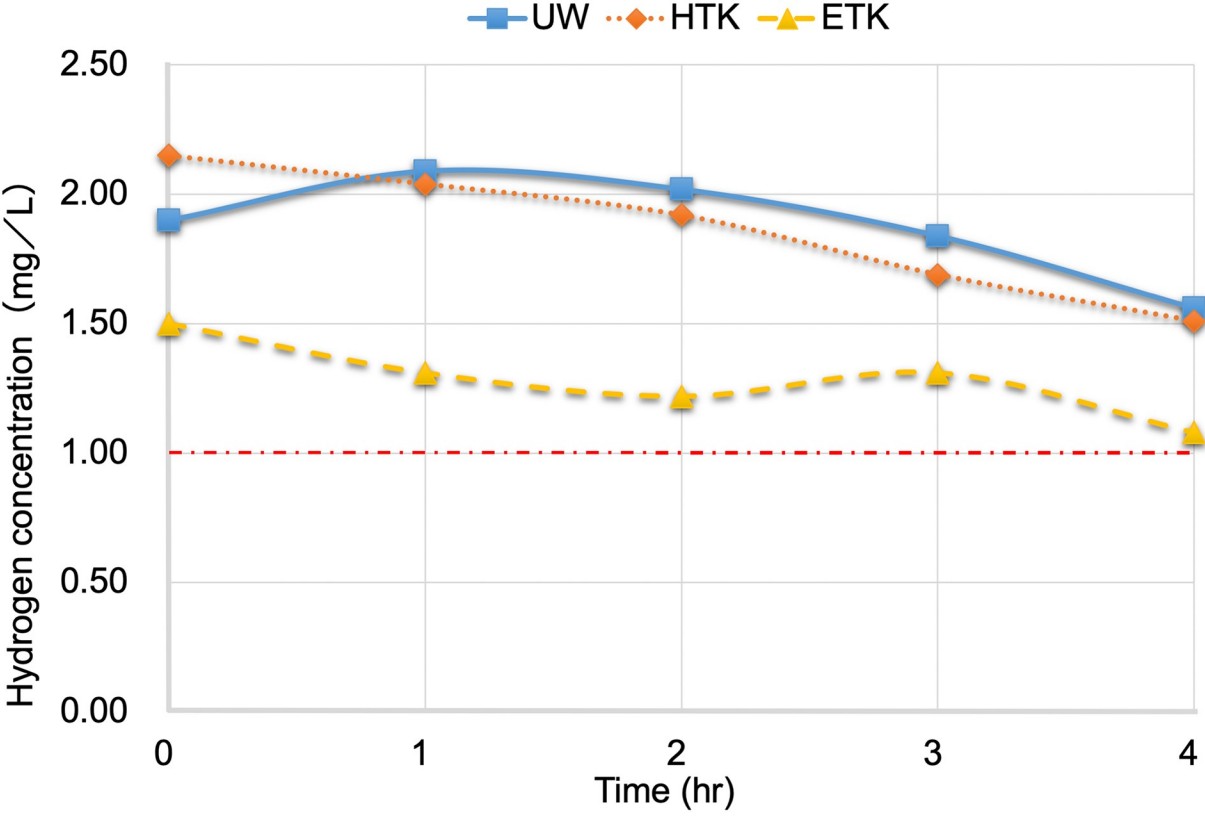

**Fig 2. Changes of the hydrogen concentration in organ preservation solutions (UW, HTK, and ETK) at atmospheric pressure after infusion of hydrogen gas.**

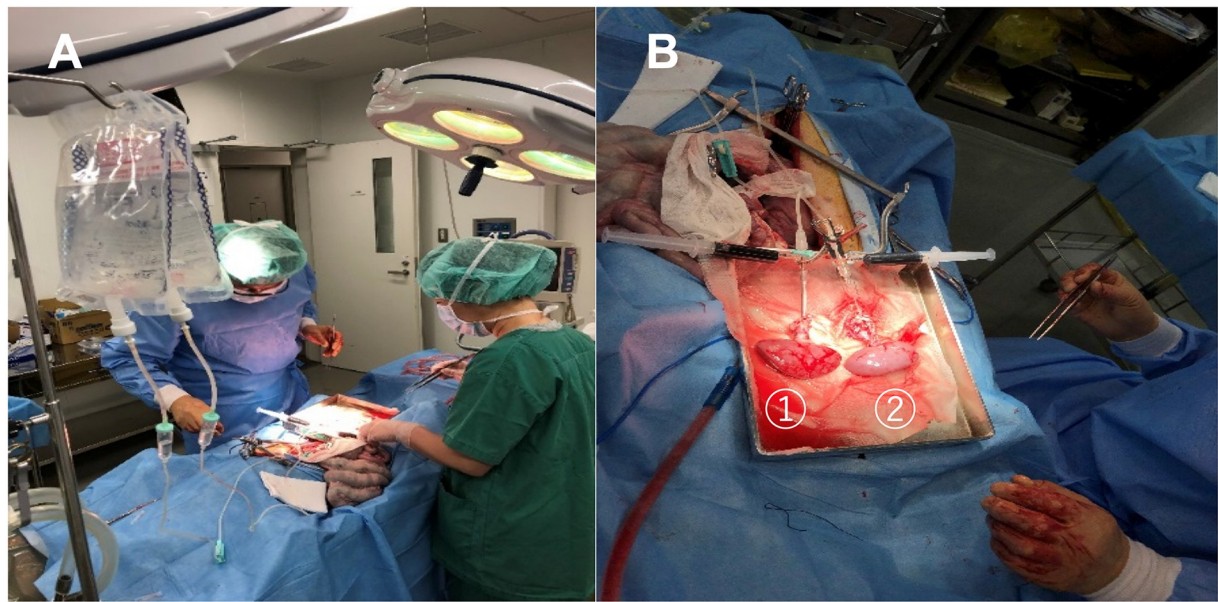

**Fig 3. Effectiveness of kidney flushing with hydrogen-containing ETK solution.** A. The kidneys were perfused for 5 minutes by gravity feed (1 m). One kidney each was perfused with hydrogen-containing ETK solution and hydrogen-free ETK solution. B. 1) Kidney perfused with hydrogen-containing ETK solution; 2) Kidney perfused with hydrogen-free ETK solution.

**Table 1. ETK solution perfusion rate by gravity feed (1 m).**

| Time (min) | Experiment 1 | | | | | Experiment 2 | | | | |
|---|---|---|---|---|---|---|---|---|---|---|
| | 0–1 | 1–2 | 2–3 | 3–4 | 4–5 | 0–1 | 1–2 | 2–3 | 3–4 | 4–5 |
| Hydrogen-containing ETK solution | 71 | 71 | 81 | 88 | 89 | 137 | 139 | 146 | 150 | 168 |
| Hydrogen-free ETK solution | 60 | 63 | 68 | 68 | 73 | 78 | 78 | 83 | 85 | 88 |

The perfusion rate is shown as drops per minute.

with hydrogen-containing ETK solution was more effective than use of hydrogen-free ETK solution (Fig 3B).

## Early post-transplant assessment of kidneys preserved with or without hydrogen-containing ETK solution

All of the recipient miniature pigs survived until the end of the observation period. After kidney storage for 1 hour before transplantation in hydrogen gas-free preservation solution, no urine was observed in the bladder and no blood flow was detected in the transplanted kidney at sacrifice on postoperative day 6 (Table 2). On the other hand, after kidney storage in hydrogen gas-containing preservation solution for either 1 hour or 4 hours, blood flow was detected in the transplanted kidney and urine was found in the bladder on postoperative day 6. After transplantation of kidneys stored for 1 hour in hydrogen gas-free ETK solution, the blood levels of BUN and creatinine in the recipients were 270 ± 29.5 mg/dL and 4.72 ± 2.72 mg/dL, respectively (n = 3). After transplantation of a single kidney stored for 1 hour in hydrogen-containing ETK solution, blood levels of BUN and creatinine were 83 mg/dL and 4.4 mg/dL (n = 1), respectively. Therefore, the kidney storage time before transplantation was extended to 4 hours for the remaining two pigs in the hydrogen-containing ETK solution group. Despite longer storage, blood levels of BUN and creatinine were low at the time of sacrifice in 1 of these 2 animals, being 108 mg/dL and 8.2 mg/dL, respectively.

Histopathological findings are displayed in Fig 4. In the hydrogen-free ETK solution group, all three kidneys showed extensive cortical necrosis (pan-necrosis). In the hydrogen gas-containing ETK solution group, there was evidence of acute tissue injury, but not cortical necrosis. Tubular dilation and cellular infiltration were noted at a low magnification, while infiltration of mononuclear cells and capillary occlusion were observed in the glomeruli at a high magnification. Infiltration of mononuclear cells and lymphocytes was also observed in the

**Table 2. Laboratory data on day 6 after kidney transplantation.**

| Group | | Storage time (hr) | Blood | | Urine | | | | | | | |
|---|---|---|---|---|---|---|---|---|---|---|---|---|
| | | | BUN (mg/dL) | CRE (mg/dL) | U-TP (mg/dL) | U-UN (mg/dL) | U-CRE (mg/dL) | U-Na (mEq/L) | U-K (mEq/L) | U-CI (mEq/L) | U-IP (mg/dL) | U-GLU (mg/dL) |
| Hydrogen-free ETK solution | No.1 | 1 | 261 | 22.8 | N.T | | | | | | | |
| | No.2 | 1 | 224 | 20.0 | N.T | | | | | | | |
| | No.3 | 1 | 325 | 29.2 | N.T | | | | | | | |
| Hydrogen-containing ETK solution | No.1 | 1 | 83 | 4.4 | 327.6 | 405.5 | 99.07 | 15 | 35.5 | 12 | 0.8 | 8 |
| | No.2 | 4 | 108 | 8.2 | 41.0 | 682.5 | 81.28 | 23 | 30.6 | 13 | 44.2 | 1 |
| | No.3 | 4 | 239 | 33.6 | 2491.0 | 290.1 | 51.11 | 89 | 39.9 | 75 | 17.9 | 61 |

N.T; Not tested (no urine in the bladder)

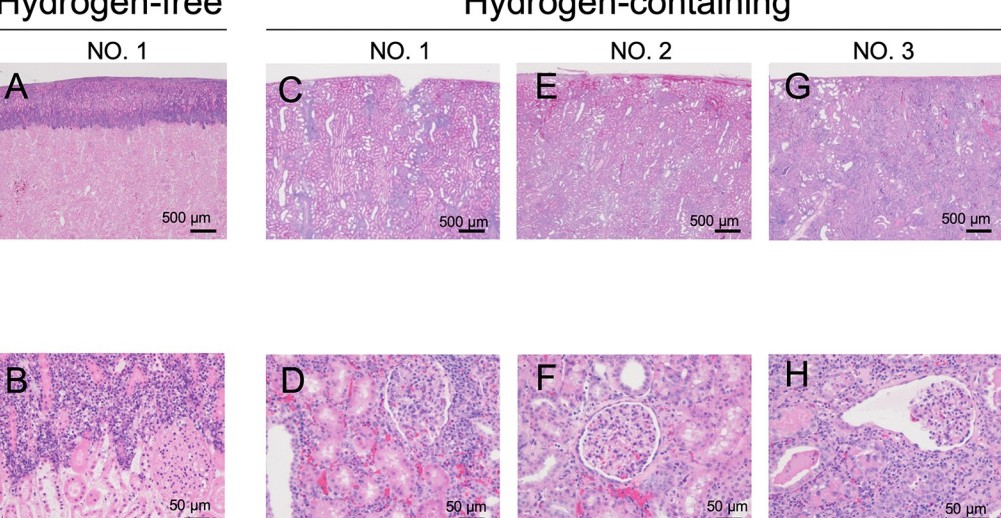

**Fig 4. Histological findings of the kidneys on day 6 after transplantation.** Specimen numbers correspond to those in Table 2. A, C, E, G: low magnification (40x), B, D, F, H: high magnification (400x). (A, B) A kidney preserved by using hydrogen-free ETK solution shows extensive necrosis in most of the microscopic field, indicating pan-necrosis of the cortex. (C-H) No cortical necrosis was observed in kidneys preserved by using hydrogen gas-containing ETK solution. However, tubular dilation and interstitial mononuclear cell infiltration are evident.

tubulointerstitial region. The pathological findings of transplanted kidneys preserved by using hydrogen-containing ETK solution are summarized in Table 3.

## Discussion

In this study, we developed a method of producing hydrogen-containing organ preservation solution by rapidly infusing hydrogen gas into organ preservation solution from a canister containing hydrogen-absorbing alloy. Since the source of hydrogen gas is a hydrogen-absorbing alloy canister, it can be easily and safely brought to the site of donor organ extraction, and hydrogen gas can be quickly infused into the organ preservation solution even in an emergency. Then the hydrogen-containing organ preservation solution can be used to flush the donor organ and preserve it during transport. When hydrogen gas was infused into a bag containing cold ETK organ preservation solution at a pressure of 0.06 MPa and the bag was subsequently opened to the air, we confirmed that the dissolved hydrogen concentration remained at 1.0 mg/L or higher for 4 hours.

While it has been shown that hydrogen gas is effective for preventing ischemia reperfusion injury [1–4], hydrogen gas is flammable at high concentrations and clinical use is limited by safety considerations. Our previous investigator-initiated clinical trial demonstrated that inhalation of hydrogen gas combined with coronary intervention could inhibit left ventricular remodeling after myocardial infarction [30]. In that trial, the subjects inhaled a gas mixture of hydrogen (1.3%), oxygen, and nitrogen. However, it is considered difficult to use a cylinder of pure hydrogen gas to dissolve hydrogen in an organ preservation solution at the site of organ donation.

Previous methods for dissolving hydrogen in organ preservation solutions have involved generation of hydrogen by electrolysis [18, 20, 21, 22] or by use of a hydrogen-generating agent [17], after which the hydrogen gas undergoes passive diffusion into the organ preservation solution. However, these methods are complicated and are unsuitable for clinical use

**Table 3. Detailed histopathological findings of transplanted kidneys preserved with hydrogen-containing ETK solution.**

| Findings/ specimen no. | 1 | 2 | 3 |
|---|---|---|---|
| **Cortical necrosis** | 0 | 0 | 0 |
| Cell debris | 0 | 0 | 0 |
| **Acute rejection** | **2** | **1** | **3** |
| Glomerulitis | 3 | 1 | 2 |
| Glomerular thrombus | 0 | 0 | 1 |
| Thrombus | 0 | 0 | 2 |
| Interstitial mononuclear cell infiltration | 3 | 2 | 3 |
| Tubulitis | 3 | 2 | 3 |
| Intimal arteritis | 2 | 0 | 3 |
| Intimal arteritis in medulla | 0 | 3[F] | 0 |
| **Chronic allograft nephropathy** | **1** | **1** | **1** |
| Chronic transplant glomerulopathy, mesangial proliferation | 2 | 1 | 2 |
| Interstitial fibrosis | 1 | 1 | 1 |
| Tubular atrophy and loss | 1 | 2 | 3 |
| Tubular degeneration and necrosis | 0 | 1 | 2 |
| Tubular dilatation | 2 | 2 | 3 |
| Tubular regeneration | 0 | 1 | 1 |
| Fibrous intimal thickening with elastica fragmentation | 0 | 2 | 3 |
| Other | | | |
| Hyaline casts | 1 | 2 | 3 |
| Cellular casts | 1 | 1 | 2 |
| Neutrophil/ eosinophil infiltration | 1 | 1 | 1 |
| Hemorrhage | 1 | 1 | 1 |
| Congestion, outer medulla | 2 | 1 | 1 |

Grading system: 0; No change, 1: Mild, 2: Moderate, 3; Severe.

[F] Fibrinoid change: nonspecific changes in arteries or veins

since it requires 24–48 hours for the hydrogen gas concentration in the solution to reach 1 mg/L.

When hydrogen-containing ETK solution was compared with hydrogen-free ETK solution for perfusion of donor kidneys, the former showed a higher flow rate from the start of perfusion. Examination of the perfused tissue revealed dilation of capillaries and washout of microthromboses.

In this study, we performed short-term observation (up to the 6th postoperative day) after transplantation of kidneys with warm ischemic injury from donor miniature pigs with circulatory arrest. When transplantation was done after storage of the kidney in hydrogen gas-free organ preservation solution for 1 hour, the outcome was always primary non-function. In contrast, when transplantation was performed after storage in hydrogen-containing solution for up to 4 hours, blood flow was detected in the transplanted kidney and production of urine was observed. These results suggest that our method could potentially be used clinically for restoring damaged organs to transplantable organs.

Some limitations of our study should be considered. We focused on the acute postoperative period, so it is also necessary to examine long-term renal function in recipients on immunosuppressant therapy in the future. In addition, kidneys treated with the hydrogen gas-containing organ preservation solution still developed severe acute tubulointerstitial injury. However,

it is possible that the combination of hydrogen-containing organ preservation solution with an intraoperative rinse [31], intraoperative or postoperative immunosuppressant therapy, or adjuvant cell therapy using cells such as mesenchymal stromal cells [32, 33] could allow kidneys from marginal donors that have not been usable to be transplanted in the future.

## Conclusion

In conclusion, we developed a device that can be used to rapidly infuse hydrogen gas into organ preservation solution at the site of donor organ removal. It is possible that flushing and preservation of organs with hydrogen gas-containing solution could increase the feasibility of performing transplantation from marginal donors.

## Supporting information

**S1 File. NC3Rs ARRIVE guidelines checklist.**
(PDF)

## Acknowledgments

This work was supported by grants from Doctors Man Co., Ltd. (E.K.). Sou Hashimoto (Doctors Man Co., Ltd.) assisted with devising the system for infusing hydrogen gas into organ preservation solution.

## Author Contributions

**Supervision:** Motoaki Sano.

**Writing – original draft:** Eiji Kobayashi, Motoaki Sano.

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
