## [Decision Letter · Decision Letter 0]

24 Jun 2019

PONE-D-19-15185

Organ preservation solution containing dissolved hydrogen gas from a hydrogen-absorbing alloy canister improves function of transplanted ischemic kidneys in miniature pigs

PLOS ONE

Dear Dr. Sano,

Thank you for submitting your manuscript to PLOS ONE. After careful consideration, we feel that it has merit but does not fully meet PLOS ONE’s publication criteria as it currently stands. Therefore, we invite you to submit a revised version of the manuscript that addresses the points raised during the review process.

We would appreciate receiving your revised manuscript by Aug 08 2019 11:59PM. To enhance the reproducibility of your results, we recommend that if applicable you deposit your laboratory protocols in protocols.io, where a protocol can be assigned its own identifier (DOI) such that it can be cited independently in the future. For instructions see: http://journals.plos.org/plosone/s/submission-guidelines#loc-laboratory-protocols

We look forward to receiving your revised manuscript.

Kind regards,

Tohru Minamino, M.D., Ph.D.

Academic Editor

PLOS ONE

Journal Requirements:

2. To comply with PLOS ONE submissions requirements, in your Methods section, please provide additional information on the animal research and ensure you have included details on (1) methods of sacrifice, (2) methods of anesthesia and/or analgesia, (3) efforts to alleviate suffering, (4) number of animals, (5) source of animals.

3. As part of your revision, please complete and submit a copy of the ARRIVE Guidelines checklist, a document that aims to improve experimental reporting and reproducibility of animal studies for purposes of post-publication data analysis and reproducibility: https://www.nc3rs.org.uk/arrive-guidelines. Please include your completed checklist as a Supporting Information file. Note that if your paper is accepted for publication, this checklist will be published as part of your article.

4. Thank you for stating the following in the Financial Disclosure section:

This work was supported by grants from Doctors Man Co., Ltd. Doctors Man Co., Ltd had no role in study design, data collection and analysis, decision to publish, or preparation of the manuscript.

We note that you received funding from a commercial source: Doctors Man Co., Ltd.

Additional Editor Comments:

We noted that certain details of animal experimental procedures are missing from the methods, specifically a description of the method of [sacrifice/euthanasia, anaesthesia, etc.].

Reviewers' comments:

Reviewer's Responses to Questions

**Comments to the Author**

1. Is the manuscript technically sound, and do the data support the conclusions?

Reviewer #1: Yes

Reviewer #2: Yes

2. Has the statistical analysis been performed appropriately and rigorously? 

Reviewer #1: Yes

Reviewer #2: N/A

3. Have the authors made all data underlying the findings in their manuscript fully available?

Reviewer #1: Yes

Reviewer #2: Yes

4. Is the manuscript presented in an intelligible fashion and written in standard English?

Reviewer #1: Yes

Reviewer #2: Yes

5. Review Comments to the Author

Reviewer #1: General Comments:

The authors showed new method of dissolving hydrogen gas in organ preservation solution. This paper was well written. However, there are several concerns to be addressed.

Comments:

1. As shown in this paper, new methods seem easy and useful to prepare before organ transplant, especially for emergency. However, there was no discussion about cost effectiveness of this method compared to conventional methods. Please discuss about this in the discussion section.

2. As authors commented in limitation, this paper did not show long-term prognosis of transplanted organ. However, many papers showed long-term effect of hydrogen gas containing solution for transplanted organs. Authors should show long-term prognosis of transplanted organ. Readers of this journal want to know about this.

3. As shown in Organ Biology 2014; 21: 150-158 and Organ Biology 2015; 22: 117-120, the effect of hydrogen gas for transplanted organ is related to anti-inflammatory function, such as inducing heme oxygenase-1 and/or activating NF-E2-related factor 2. Authors should show the molecular mechanism of hydrogen-gas containing solution for transplanted kidney in this method.

4. There was no data about the biological significance of hydrogen gas containing ETK solution compared to UW and HTK. As authors declare UW and HTK solution had high hydrogen concentration at the beginning, authors should show the biologically significant data about ETK.

5. As authors commented in limitation, it is well known that hydrogen-gas containing solutions still develop severe tubulointerstitial injury. However, in this study, transplanted kidney immediately worked well after transplanted. Is there any biological advantage in this new method? Authors should show the effectiveness compared to other hydrogen-gas containing methods.

Minor comments:

none

Reviewer #2: They developed a method of producing hydrogen-containing organ preservation solution by rapidly infusing hydrogen gas into organ preservation solution from a canister containing hydrogen-absorbing alloy. The results are interesting. If they can perform additional histology of azan staining or tunel staining, it would be more convincing.

6. PLOS authors have the option to publish the peer review history of their article (what does this mean?). If published, this will include your full peer review and any attached files.

Reviewer #1: Yes: Yuichiro Maekawa

Reviewer #2: No

---

## [Author Response · Author response to Decision Letter 0]

29 Aug 2019

Thank you for the constructive suggestions for improving our original manuscript. According to the suggestions, we have revised the manuscript. Below please find our reply to each comment you kindly made in a point-by-point manner (your points in italics and our responses in plain type).

Point-by-point reply to Ms. Vicky Stabler

Reply: We confirmed that our manuscript meets PLOS ONE's style requirements.

2. To comply with PLOS ONE submissions requirements, in your Methods section, please provide additional information on the animal research and ensure you have included details on (1) methods of sacrifice, (2) methods of anesthesia and/or analgesia, (3) efforts to alleviate suffering, (4) number of animals, (5) source of animals.

Reply: Information on (1) methods of sacrifice, (2) methods of anesthesia and/or analgesia, (3) efforts to alleviate suffering, (4) number of animals, (5) source of animals, were added to a “Miniature pigs” paragraph in the Material and Methods section as described below.

We used female pigs aged 25-40 months and weighing 20-26 Kg, purchased from Fujimicra Ltd., Shizuoka, Japan. Animals were treated in accordance with the Animal (Scientific Procedures) Protection Act 1986 of the United Kingdom. The pigs were housed in cages under temperature and light-controlled conditions (12-hour light/dark cycle) and were provided with food and water ad libitum. The pigs were fasted for 12 hours prior to surgery with free access to water. Sedation with a mixture of midazolam/medetomidine/butorphanol was followed by endotracheal intubation and mechanical ventilation. Anesthesia was maintained with inhalational isoflurane. Midazolam and medetomidine were added according to the depth of anesthesia. Buprenorphine was administered as an intraoperative analgesic management. After intravenous administration of pentobarbital, saturated potassium chloride was rapidly administered intravenously to euthanize. Three animals were used to examine the flushing effect of the hydrogen-containing organ preservation solution and nine animals were used for the organ transplantation experiment (one donor and two recipients were used for one set of experiments; n = 3). These experiments were conducted with the approval of the Research Council and Animal Care and Use Committee of Keio University [approval no: 16072-(1)]. Surgery was performed by a surgeon with over 200 clinical transplant operations, who is a steering member of the transplantation society and a permanent director of the transplantation society of Japan (E.K.).

3. As part of your revision, please complete and submit a copy of the ARRIVE Guidelines checklist, a document that aims to improve experimental reporting and reproducibility of animal studies for purposes of post-publication data analysis and reproducibility: https://www.nc3rs.org.uk/arrive-guidelines. Please include your completed checklist as a Supporting Information file. Note that if your paper is accepted for publication, this checklist will be published as part of your article.

Reply: We uploaded a copy of the ARRIVE Guidelines checklist as a Supporting Information file.

4. Thank you for stating the following in the Financial Disclosure section: This work was supported by grants from Doctors Man Co., Ltd. Doctors Man Co., Ltd had no role in study design, data collection and analysis, decision to publish, or preparation of the manuscript. We note that you received funding from a commercial source: Doctors Man Co., Ltd. Please provide an amended Competing Interests Statement that explicitly states this commercial funder, along with any other relevant declarations relating to employment, consultancy, patents, products in development, marketed products, etc. Within this Competing Interests Statement, please confirm that this does not alter your adherence to all PLOS ONE policies on sharing data and materials by including the following statement: "This does not alter our adherence to PLOS ONE policies on sharing data and materials.” (as detailed online in our guide for authors http://journals.plos.org/plosone/s/competing-interests). If there are restrictions on sharing of data and/or materials, please state these. Please note that we cannot proceed with consideration of your article until this information has been declared. Please include your amended Competing Interests Statement within your cover letter. We will change the online submission form on your behalf.

Reply: We amended Competing Interests Statement and included our amended Competing Interests Statement within our cover letter.

 

Point-by-point reply to the editors and reviewers

Editors

We noted that certain details of animal experimental procedures are missing from the methods, specifically a description of the method of [sacrifice/euthanasia, anaesthesia, etc.].

Reply: Information on (1) methods of sacrifice, (2) methods of anesthesia and/or analgesia, (3) efforts to alleviate suffering, (4) number of animals, (5) source of animals, were added to a “Miniature pigs” paragraph in the Material and Methods section. 

We used female pigs aged 25-40 months and weighing 20-26 Kg, purchased from Fujimicra Ltd., Shizuoka, Japan. Animals were treated in accordance with the Animal (Scientific Procedures) Protection Act 1986 of the United Kingdom. The pigs were housed in cages under temperature and light-controlled conditions (12-hour light/dark cycle) and were provided with food and water ad libitum. The pigs were fasted for 12 hours prior to surgery with free access to water. Sedation with a mixture of midazolam/medetomidine/butorphanol was followed by endotracheal intubation and mechanical ventilation. Anesthesia was maintained with inhalational isoflurane. Midazolam and medetomidine were added according to the depth of anesthesia. Buprenorphine was administered as an intraoperative analgesic management. After intravenous administration of pentobarbital, saturated potassium chloride was rapidly administered intravenously to euthanize. Three pigs were used to examine the flushing effect of the hydrogen-containing organ preservation solution and nine pigs were used for the organ transplantation experiment (one donor and two recipients were used for one set of experiments; n = 3). These experiments were conducted with the approval of the Research Council and Animal Care and Use Committee of Keio University [approval no: 16072-(1)]. Surgery was performed by a surgeon with over 200 clinical transplant operations, who is a steering member of the transplantation society and a permanent director of the transplantation society of Japan (E.K.).

Animal experimental procedures for donor surgery and recipient surgery are detailed in “Investigation of the effectiveness of hydrogen-containing preservation solution for flushing kidneys from donor pigs with circulatory arrest” and “Evaluation in a kidney transplantation model” paragraph in Materials and Methods session.

Below, we described separately for Donor surgery and Recipient surgery.

Donor surgery

A midline upper abdominal incision was made under general anesthesia, and the left and right kidneys were dissected. Then the chest was opened and the thoracic aorta was clamped to induce ischemia of all abdominal organs. After 30 minutes of warm ischemia, the kidneys were removed for use as donor organs. The donor animal was not treated with an anticoagulant such as heparin. The excised kidneys were immediately placed on the back table, divided into the right and left kidneys, and perfused for 5 minutes by gravity feed from a height of 1 m. One kidney was perfused with hydrogen-containing cold extracellular-type trehalose-containing Kyoto (ETK) solution and the other was perfused with hydrogen-free cold ETK solution. The number of drops of the solution per minute was counted for each kidney. 

Recipient surgery

Under general anesthesia, a midline abdominal incision was made to expose the recipient’s left renal artery and vein (24). After intravenous administration of 1 ml of heparin, the abdominal aorta was clamped immediately above the renal artery bifurcation. Then the renal vein was clamped with bulldog forceps, and the left kidney was resected from the renal artery leaving a Carrel patch configuration. Next, the stored donor kidney was anastomosed end-to-end with 5-0 nylon to the renal artery at the Carrel patch site. After arterial anastomosis, the peripheral renal artery was clipped again before unclamping the abdominal aorta. The total aortic clamp time was 30 minutes. Subsequently, the renal veins were joined by end-to-end continuous anastomosis with 6-0 nylon, and finally the ureter was anastomosed with 6-0 nylon knotted sutures. The transplanted kidney was wrapped in a thermal bag, and the temperature was kept at 20 °C or less by appropriate infusion of cold ETK solution (25). Warm ischemic time was set at 1 hour. After reperfusion, blood flow was confirmed in the transplanted kidney. Then the right native kidney was excised and the abdomen was closed. Cefazolin sodium (25mg/kg) was given on the first postoperative day, and the animals were allowed free access to both food and water. The animals were observed for up to 6 days after surgery, and then were sacrificed for laboratory tests, urinalysis, and histopathological examination of the transplanted kidneys.

 

Reviewer #1 

1. As shown in this paper, new methods seem easy and useful to prepare before organ transplant, especially for emergency. However, there was no discussion about cost effectiveness of this method compared to conventional methods. Please discuss about this in the discussion section.

Response:

Thank you very much for kind remarks and thoughtful comments. For drinking water, a hydrogen-absorbing alloy canister is sold at a rental fee of ¥ 30,000 per month. This technology has been patented and we want to make it extremely cheap to effectively spread. The superiority compared to other devices is described in the discussion.

2. As authors commented in limitation, this paper did not show long-term prognosis of transplanted organ. However, many papers showed long-term effect of hydrogen gas containing solution for transplanted organs. Authors should show long-term prognosis of transplanted organ. Readers of this journal want to know about this.

Response

The use of immunosuppressants is necessary to achieve long-term results. The paper was done without the use of immunosuppressive agents to see the direct effects of hydrogen gas alone. In the future, a long-term effect of hydrogen gas containing solution for transplanted organs using immunosuppressive drugs is also planned.

3. As shown in Organ Biology 2014; 21: 150-158 and Organ Biology 2015; 22: 117-120, the effect of hydrogen gas for transplanted organ is related to anti-inflammatory function, such as inducing heme oxygenase-1 and/or activating NF-E2-related factor 2. Authors should show the molecular mechanism of hydrogen-gas containing solution for transplanted kidney in this method.

Response 

Because it is a very marginal graft in a large animal model, a biopsy immediately after reperfusion was avoided. In this paper, we have shown clinical utility, not a mechanism.

4. There was no data about the biological significance of hydrogen gas containing ETK solution compared to UW and HTK. As authors declare UW and HTK solution had high hydrogen concentration at the beginning, authors should show the biologically significant data about ETK.

Response

In the previous experiment on transplantation of organs removed from rats after cardiac arrest, we observed that edema was reduced by using extracellular-type trehalose containing Kyoto solution (ETK) as a flash-out solution. As a result, ETK was used. References were as follows.

1. Iwai S, Kikuchi T, Kasahara N, Teratani T, Yokoo T, Sakonju I, et al. Impact of normothermic preservation with extracellular type solution containing trehalose on rat kidney grafting from a cardiac death donor. PLoS One 2012; 7: e33157 (Ref 28) 

2. Kaimori JY, Iwai S, Hatanaka M, Teratani T, Obi Y, Tsuda H, Isaka Y, Yokawa T, Kuroda K, Ichimaru N, Okumi M, Yazawa K, Rakugi H, Nonomura N, Takahara S, Kobayashi E. Non-invasive magnetic resonance imaging in rats for prediction of the fate of grafted kidneys from cardiac death donors. PLoS One. 2013 May 7;8(5):e63573 (Added as Ref 33)

5. As authors commented in limitation, it is well known that hydrogen-gas containing solutions still develop severe tubulointerstitial injury. However, in this study, transplanted kidney immediately worked well after transplanted. Is there any biological advantage in this new method? Authors should show the effectiveness compared to other hydrogen-gas containing methods.

Response

Thank you very much for kind remarks. The biological advantage of hydrogen-gas containing organ preservation solution for transplanted organs has been proven in many studies so far (Ref 6-22). In this study, we showed that hydrogen-gas containing organ preservation solution can revive damaged organs removed from an elderly miniature pig whose blood flow has stopped for 30 minutes to a transplantable organ to some extent. Various methods have been devised to dissolve hydrogen gas in organ preservation solutions, including use of a hydrogen gas cylinder, electrolysis, or a hydrogen-generating agent. However, these methods require considerable time and effort for preparation. We first made a practical hydrogen gas supply device at the site where organs are removed from donors. The canister storing hydrogen can be safely transported anywhere and can be easily connected to a bag containing conventional organ preservation solution, allowing hydrogen to be dissolved in organ preservation solution within a few minutes at the site of donor organ harvesting. These advantages are described in the discussion.

Reviewer #2: 

They developed a method of producing hydrogen-containing organ preservation solution by rapidly infusing hydrogen gas into organ preservation solution from a canister containing hydrogen-absorbing alloy. The results are interesting. If they can perform additional histology of azan staining or tunel staining, it would be more convincing.

Response

Thank you very much for a kind remark and a suggestion. The kidney sessions were assessed based on Banff classification of Renal Allograft Pathology, an international consensus on renal transplant biopsy reporting (Table 3).

According to reviewer’s suggestion, TUNEL staining was performed, and the number of TUNEL positive cells in tubular epithelial cells was counted. There was a tendency for the number of TUNEL positive cells to be less with hydrogen-containing solution compared with hydrogen-free solution under the condition of 1 hour storage time, but the difference was not significant. At 6 days after transplantation, with hydrogen-containing solution, the increase in BUN/CRTNIN was suppressed, urine excretion was observed, and histologically, transplanted kidney did not fall into cortical necrosis. By contrast, sub-optimal (marginal) kidney did not tolerate conventional cold storage and showed cortical necrosis and never function (primary non-functioning).

These results suggest that hydrogen may suppress cell death by mechanisms other than apoptosis.　Or as a more likely possibility, hydrogen gas could restore the microvasculature and improve the blood circulation, thereby accelerating the functional recovery of the transplanted kidney, rather than suppressing apoptotic cell death based on ischemia-reperfusion injury. This is consistent with the result that hydrogen-containing solution promotes the effect of washing out blood from organs extracted from marginal donors. Examination of the perfused tissue revealed dilation of capillaries and washout of micro-thromboses. Since this content is related to the patent currently being prepared for application, we will refrain from making a description in this paper.

---

## [Decision Letter · Decision Letter 1]

10 Sep 2019

[EXSCINDED]

Organ preservation solution containing dissolved hydrogen gas from a hydrogen-absorbing alloy canister improves function of transplanted ischemic kidneys in miniature pigs

PONE-D-19-15185R1

Dear Dr. Sano,

We are pleased to inform you that your manuscript has been judged scientifically suitable for publication and will be formally accepted for publication once it complies with all outstanding technical requirements.

With kind regards,

Tohru Minamino, M.D., Ph.D.

Academic Editor

PLOS ONE

Additional Editor Comments (optional):

Reviewers' comments:

Reviewer's Responses to Questions

**Comments to the Author**

1. If the authors have adequately addressed your comments raised in a previous round of review and you feel that this manuscript is now acceptable for publication, you may indicate that here to bypass the “Comments to the Author” section, enter your conflict of interest statement in the “Confidential to Editor” section, and submit your "Accept" recommendation.

Reviewer #1: All comments have been addressed

Reviewer #2: All comments have been addressed

2. Is the manuscript technically sound, and do the data support the conclusions?

Reviewer #1: Partly

Reviewer #2: Yes

3. Has the statistical analysis been performed appropriately and rigorously? 

Reviewer #1: Yes

Reviewer #2: N/A

4. Have the authors made all data underlying the findings in their manuscript fully available?

Reviewer #1: Yes

Reviewer #2: Yes

5. Is the manuscript presented in an intelligible fashion and written in standard English?

Reviewer #1: No

Reviewer #2: Yes

6. Review Comments to the Author

Reviewer #1: The manuscript is adequately corrected. However, English editing is required by native English speaker before publication.

Reviewer #2: The authors performed additonal experiments and answered well to my concerns. This paper is now acceptable.

7. PLOS authors have the option to publish the peer review history of their article (what does this mean?). If published, this will include your full peer review and any attached files.

Reviewer #1: No

Reviewer #2: No

---

## [Editor Report · Acceptance letter]

23 Sep 2019

PONE-D-19-15185R1 

Organ preservation solution containing dissolved hydrogen gas from a hydrogen-absorbing alloy canister improves function of transplanted ischemic kidneys in miniature pigs 

Dear Dr. Sano:

I am pleased to inform you that your manuscript has been deemed suitable for publication in PLOS ONE. Congratulations! Your manuscript is now with our production department. 

With kind regards,

on behalf of

Professor Tohru Minamino 

Academic Editor

PLOS ONE